# Triangular Functionally Graded Porous Moderately Thick Plates—Deformations and Free Vibrations

**Aleksander Muc**

Department of Physics, Faculty of Materials Engineering and Physics, Cracow University of Technology, 31-155 Kraków, Poland; olekmuc@mech.pk.edu.pl

**Abstract:** Since the finite element analysis of deformations for plates made of functionally graded materials cannot be carried out with the use of commercial FE packages, aconsistent method of analytical analysis is proposed in the paper. The method of the analysis is based on the application of the weighted residuals and the Bubnov–Galerkin method. The 2D formulation of moderately thick plates is adopted herein for classical and transverse shear deformation plate models (first and third order). Plate deformations subjected to uniform normalpressure and free vibrations are considered. The validity of the analytical model was verified by the comparison of results with FE analysis for isotropic plates. Two correction multipliers were proposed in order to take into account the unsymmetric composition of functionally graded porous plate walls.

**Keywords:** triangular plates; functionally graded materials; deformations; free vibrations

## 1. Introduction

Triangular plates are widely used in different applications in many industrial and engineering fields (e.g., mechanical, aerospace, automotive, etc.). In many cases, the rapid and efficient evaluation of the deformations, the natural vibration frequencies, and associated mode shapes is fundamental in their design.

Due to more complicated forms of triangles and/or polygons (various partial restrictions along the edges that should be included in a significant group of practical problems) the classical methods of solutions for boundary value problems of rectangular plates or shells cannot be applied and transferred easily in order to conduct the analysis for triangular plated structures. Approximate analytical methods present adifficulty regardingthe construction of simple and adequate approximation functions that can be applied to the entire domain of the plate (since the mathematical structure of the boundary conditions becomes complex and the generation of approximating functions becomes very difficult).

In addition, it is also important to note that porous functionally graded materials (FGM) are characterized by a smooth variation of the properties from the bottom to the top surface. The classical commercial finite element codes (e.g., Abaqus, Ansys) do not possess the appropriate type of finite elements that can be adopted to the analysis of structures made of functionally graded materials.

The mentioned limitations of the present analysis are demonstrated in the review papers [1,2]. For triangular plates, an excellent source of reference are the works of Leissa [3–6].

The methods of computations of deformations and natural frequencies are classical and similar to those presented in the pioneering works of Leissa [3–6]. More information is demonstrated in Section 3.

The fundamental innovations and significanceof the present work areas follows:

- To illustrate that for FGM triangular plates derived with the use of classical Love-Kirchhoff plate theory (CPT), the influence of material properties can be expressed by two multipliers ($M$ and $\hat{M}$) similarly as it was verified for FGM rectangular plates (see Muc, Flis [2,7]);

- The validity of the method of computations used herein was verified for isotropic structures only (Section 4) since in the open literature I could not find the results for triangular plates made of functionally graded materials.

However, it should be pointed out that the identical remarks cannot be easily transferred for plates made of unidirectional laminates or plates reinforced with nanostructures (Muc, Muc-Wierzgoń [8]) and similarly for spherical FG shells (Flis, Muc [9]).

## 2. Formulation of the Problem

Let us consider deformations of triangular plates shown in Figure 1.The plate is made of porous FGM, having either a symmetric (Figure 2a,b) or an unsymmetric pores configuration(Figure 2c)

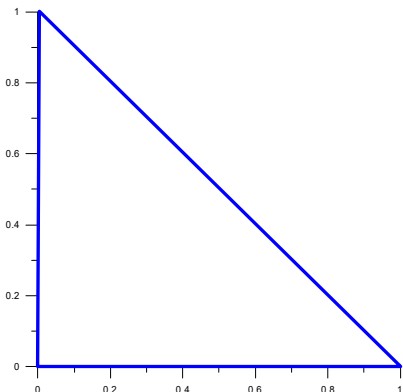

**Figure 1.** Geometry of the analyzed normalized right plates.

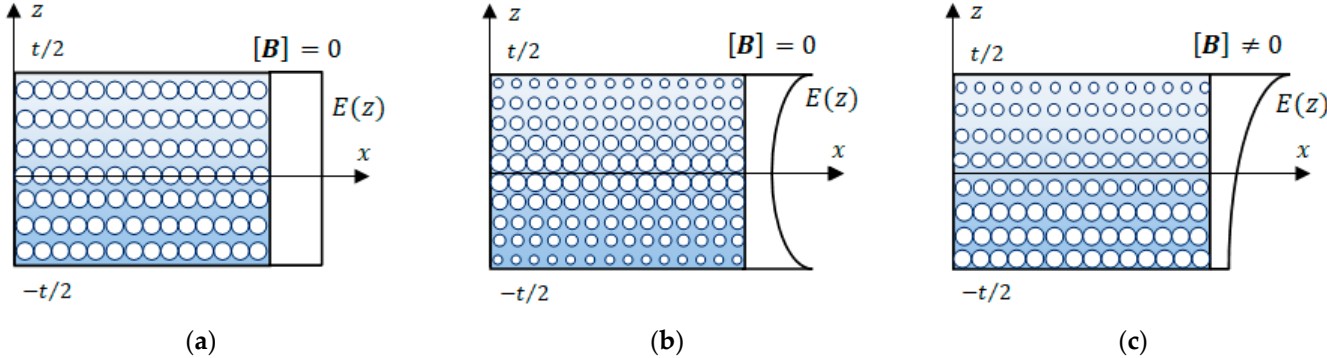

**Figure 2.** Plate wall construction for the uniform thickness $t(x,y,z) = t$: (**a**) uniform distribution of pores, (**b**) symmetric distribution, and (**c**) unsymmetric distributions.

### 2.1. Kinematic Relations

The 3-D displacements $\widetilde{U}_i(x,y,z)$ ($i$ = 1,2,3) at any point of the plate in the $x$, $y$, and $z$ directions (see also Figure 1), respectively, are expressed in the following form:

$$\widetilde{U}_1(x,y,z) = u(x,y) - z\frac{\partial w}{\partial x} + F(z)\psi_1(x,y)$$

$$\widetilde{U}_2(x,y,z) = v(x,y) - z\frac{\partial w}{\partial y} + F(z)\psi_2(x,y) \tag{1}$$

$$\widetilde{U}_3(x,y,z) = w(x,y)$$

where $u, v, w$ are the displacements of a generic point on the reference mid-surface, $\psi_1$, $\psi_2$ are the rotations of normal to the mid-surface about the $y$- and $x$-axes, respectively. The explicit form of the function $F(z)$ is presented in Table 1.

**Table 1.** Variants of plate theories.

| | Classical Plate Theory (CPT) | First Order Shear Deformation Theory (FSDT) | Higher Order Shear Deformation Theory (HSDT)—Reddy [10] | Higher Order Shear Deformation Theory (HSDT)—Hosseini-Hashemi [11] |
|---|---|---|---|---|
| $F(z)$ | 0 | $z$; the second term in Equation(1) is eliminated | $z\left(1 - \frac{4}{3}\frac{z^2}{t^2}\right)$ | $z\exp\left(-\frac{2z^2}{t^2}\right)$ |

Where $t$ denotes the total plate thickness.

The distributions of the transverse shear corrections functions $F(z)$ used in the literature [10–13] are demonstrated for first and third order transverse shear deformation theory. Let us note that for HSDT there is no need to use shear correction factors since the form of the functions leads to the parabolic distributions of transverse shear stresses. The above relations are valid for moderately thick FG plates (i.e., as the thickness, t, to the length of the plate ratio is equal to 0.05–0.1). For thick FG plates it is necessary to take into account deformations of the normal to the plate mid-surface—see [2].

Based on the above expressions, the total, linear 3-D strain tensor can be written as follows:

$$\varepsilon_{ij}(x,y,z) = \frac{1}{2}\left(\frac{\partial \widetilde{U}_i}{\partial x_j} + \frac{\partial \widetilde{U}_j}{\partial x_i}\right), x_1 = x, x_2 = y, x_3 = z, i,j = 1,2,3 \tag{2}$$

### 2.2. Constitutive Equations and Variational Formulation

Eliminating from the considerations the normal strains $\varepsilon_{33}(x,y,z)$ and stresses $\sigma_{33}(x,y,z)$, the constitutive stress-strain relations can be expressed in the following way (the Kelvin-Voigt notation) [14]:

$$\sigma_i(x,y,z) = C_{ij}\varepsilon_j(x,y,z), i,j = 1,2,\ldots 5 \tag{3}$$

where,

$$[C] = \frac{E(z)}{1-v^2}\begin{bmatrix} 1 & v & 0 & 0 & 0 \\ v & 1 & 0 & 0 & 0 \\ 0 & 0 & 0.5(1-v) & 0 & 0 \\ 0 & 0 & 0 & 0.5(1-v) & 0 \\ 0 & 0 & 0 & 0 & 0.5(1-v) \end{bmatrix}, \tag{4}$$

Young's modulus, $E$, varies through thickness direction z and Poission's ratio, is assumed as constant (see Leissa [15]).

The control of global properties (for instance, density $\rho$ and elastic modulus $E$ variations) are described by the identical function $P(z)$ characterizing the pore distributions (Figure 2):

$$P(z)/P_b = [(P_t/P_b - 1)f(z) + 1](1 - \varphi(z)) \tag{5}$$

$$f(z) = \left(\frac{z}{t} + \frac{1}{2}\right)^n,$$

Type 1: $\varphi(z) = c_p\cos\left(\frac{\pi z}{t}\right)$, Type 2: $\varphi(z) = c_p\cos\left[\frac{\pi}{2}\left(\frac{z}{t} + 0.5\right)\right]$, Type 3: $\varphi(z) = c_p\cos\left[\frac{\pi}{2}\left(\frac{z}{t} - 0.5\right)\right]$

where the symbols $t$ and $b$ refer to the material properties on the top and bottom surfaces, $n$ is the power index, $\varphi(z)$ is the distribution of porosity along the thickness direction z, and $c_p$ denotes the chosen variation of porosity.

### 2.3. Governing Relations

Governing relations are derived with the method discussed by Muc [16]. For 3D structures, let us consider the following functional:

$$H = \int_{\tau_1}^{\tau_2} (K - \Pi) d\Omega d\tau = 0 \tag{6}$$

where: $K$ denotes the total kinetic energy, $\Pi$ is the total potential energy and $\tau$ means the physical time. The total kinetic energy can be written in the following form:

$$K = \frac{1}{2} \int_\Omega \rho(z) \frac{d\widetilde{U}_i}{d\tau} \frac{d\widetilde{U}_i}{d\tau} d\Omega \tag{7}$$

and the total potential energy is defined as follows:

$$\Pi(\widetilde{U}_i) = \int_\Omega W d\Omega - \int_{S_T} p\widetilde{U}_3 dS = \Pi_{\text{int}} - \int_{S_T} p\widetilde{U}_3 dS \tag{8}$$

where $W$ is the potential of internal forces (the strain energy density function):

$$W = \frac{1}{2}\sigma_{\alpha\beta}\varepsilon_{\alpha\beta} = \frac{1}{2}(\sigma_{11}\varepsilon_{11} + \sigma_{22}\varepsilon_{22} + \sigma_{33}\varepsilon_{33}) + \sigma_{12}\varepsilon_{12} + \sigma_{13}\varepsilon_{13} + \sigma_{23}\varepsilon_{23} \tag{9}$$

$\Omega$ defines the volume occupied by the deformable body. $p$ is the uniform normal pressure in the global reference system. $S_T$ is the portion of the surface on which the pressure is specified. $\rho(z)$ is the density distribution defined with the use of the relation (5).

According to the classical macro-mechanical approach for composite structures (the homogenization of ply properties), each component of the functional in Equation (8) is evaluated as the sum of contributions from FG material.

Through integrating along the plate thickness z the stiffness matrix components are defined as follows:

$$[\mathbf{A}] = \int_{-t/2}^{t/2} [\mathbf{C}]dz, \ [\mathbf{B}] = \int_{-t/2}^{t/2} [\mathbf{C}]zdz, \ [\mathbf{D}] = \int_{-t/2}^{t/2} [\mathbf{C}]z^2dz, \ [\mathbf{F}] = \int_{-t/2}^{t/2} [\mathbf{C}]z^3dz, \ [\mathbf{H}] = \int_{-t/2}^{t/2} [\mathbf{C}]z^4d, z,$$

$$[\mathbf{I}] = \int_{-t/2}^{t/2} [\mathbf{C}]z^5dz, \ [\mathbf{J}] = \int_{-t/2}^{t/2} [\mathbf{C}] \, z^6dz \tag{10}$$

The terms $[\mathbf{F}]$, $[\mathbf{H}]$, $[\mathbf{I}]$, $[\mathbf{J}]$ characterize the influence of the higher order transverse shear deformation theories. The material data in the matrix $[\mathbf{C}]$ refer to the principal material coordinates.

Using Hamilton's Principle

$$\int_{\tau_1}^{\tau_2} \delta(K - \Pi) = 0 \tag{11}$$

one can derive the fundamental system of equations describing the deformations of the plate.

Let us notice that the above operations can be carried out in a symbolic way searching for the possible variations of unknown functions $u$, $v$, $w$, $\psi_1$, $\psi_2$ with the aid of the operation "*Variational Calculus*" that can be found in the package Mathematica (the single command *Euler Equations*). Finally, the variations of the Hamilton functional (11) leads to five differential equations. The first two characterize the in-plane effects and are functions of in-plane displacements u and v. The next three describe bending effects and are directly

connected with the description of the bending and free vibrations effects. The final system of equations is expressed in the following way:

$$L_{ij} \begin{bmatrix} u \\ v \\ w \\ \psi_1 \\ \psi_2 \end{bmatrix} = \begin{bmatrix} 0 \\ 0 \\ pw - \hat{\rho}\omega^2 w \\ 0 \\ 0 \end{bmatrix}, i, j = 1, 2, \ldots, 5 \, \hat{\rho} = \int\limits_{-t/2}^{t/2} \rho(z) dz \tag{12}$$

The symbols $L_{ij}$ denote linear differential operators obtained with the use of the Mathematica package. For FSDT, the explicit form of the operators is presented in Ref. [16].

Two of the frequencies being the solution of free vibrations relations are much higher than the third one. They are referred toas the motion of thickness-shear and thickness-twist, respectively. The primarily lateral mode corresponds to the lowest mode. Therefore, simplified frequency equations which neglect the effects of the rotatory inertia are written in Equation (12).

## 3. Method of the Solution

In the literature various numerical methods were combined for predicting the behaviors of FGM plates. A broad review of them is presented (e.g., in Refs [17,18]). The authors of Refs [17,18] proposed novel numerical methods for the analysis of FGM and piezoelectric FGM structure (square plates, skew plates, and toruses).

In the present paper the system of differential Equation (12) is considered as the set of linear equations with five unknowns ($u$, $v$, $w$, $\psi_1$, $\psi_2$) and constants $L_{ij}$. Using the classical Kramer method forsolving linear equations, the system of Equation (12) is reduced to one linear differential variable with respect to normal deflection w. The broader discussion of those problems is presented by Muc, Flis [19]. Since for HSDT and FSDT the final form of the above relation is too complex, we demonstrate the result for CPT:

$$\nabla^4 w = -pM - \omega^2 \hat{M}, \nabla^4 = \nabla^2 \nabla^2, \nabla^2 = \frac{\partial^2}{\partial x^2} + \frac{\partial^2}{\partial y^2} M = \frac{A}{-B^2 + AD}, \hat{M} = \hat{\rho} M \tag{13}$$

For thin-walled plates, the influence of material properties is described by two parameters, with $M$ and $\hat{M}$ being the function of the material properties on top and bottom surfaces, power index $n$, the distribution of porosity along the thickness direction $\varphi(z)$ and the chosen variation of porosity $c_p$ (see Equation (5)). The variations of the multipliers with the parameters characterizing the porosity are illustrated in Figure 3.

As it may be seen in Figure 3a–d the influence of porosity is similar for different parameters $c_p$ and the $P_t/P_b$ ratio. Thisdemonstrates that the description of the coupling effects via the correction factors $M$ and $\hat{M}$ is general.

For uniform or symmetric distribution of pores (Figure 2a,b B = 0) the relation (13)is reduced to the case of isotropic structures. Let us note that equating M to $1/D_0$ Equation (13) is equivalent to the formulation of isotropic plates. Thus, computing numerically with the help of finite elements deformations or eigenvalues of polygons with the bending stiffness $D_0$ one can evaluate the appropriate values for structures with non-zero values of the B matrix components.

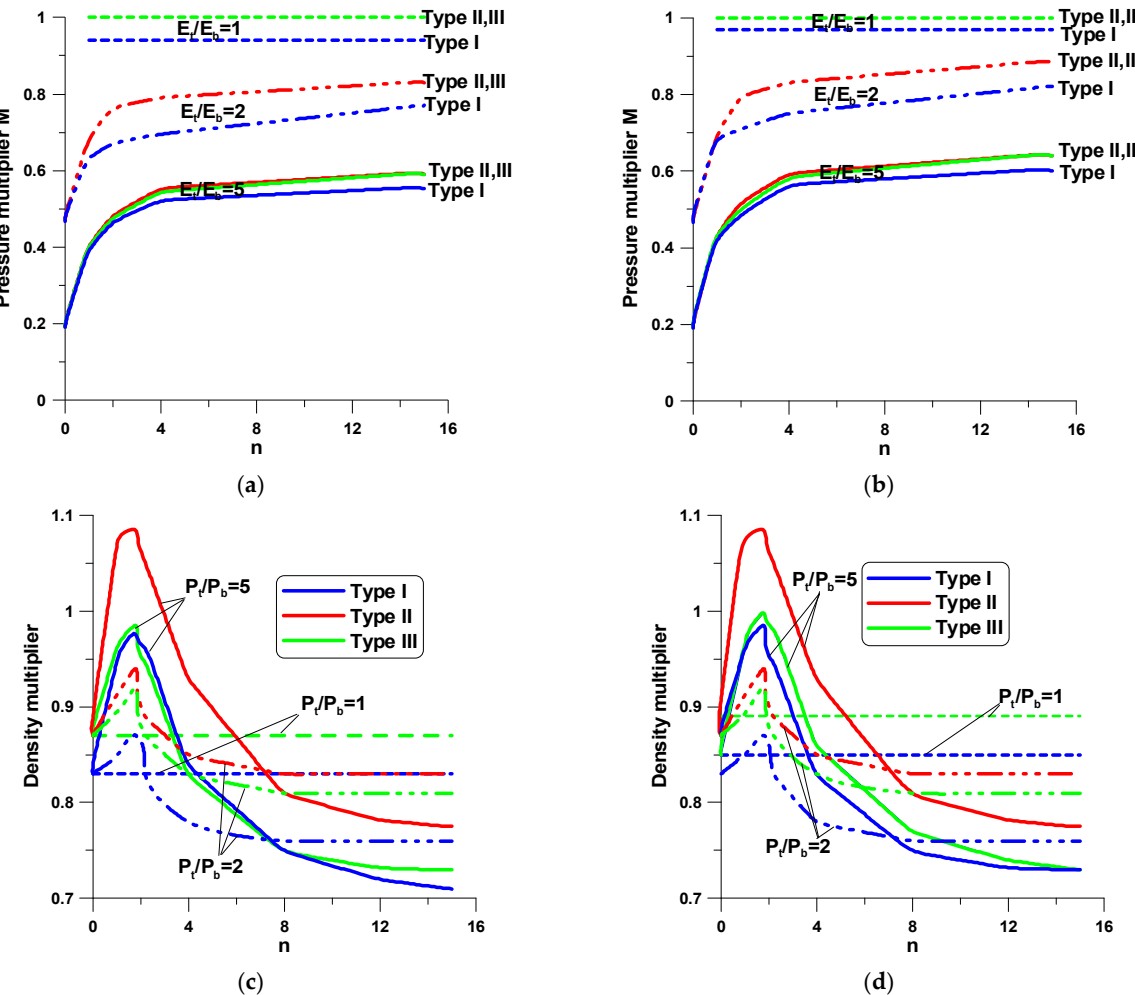

**Figure 3.** Distributions of correction multipliers and three types of porosity distributions. (**a**) Pressure multiplier $M-c_p = 0.2$. (**b**) Pressure multiplier $M-c_p = 0.5$. (**c**) natural frequencies multiplier $\hat{M}-c_p = 0.2$. (**d**) Natural frequencies multiplier $\hat{M}-c_p = 0.5$.

Different approximation techniques exist that areused to solve deformations of plates (see Leissa [3]) due to the complex form of governing equations in the form of Equation (13) (valid for FG CPT) or much more complicated for FSDT. For HSDT FG plates, the method of weighted residuals is adopted. The basic idea of the method of the weighted residuals is to use a trial function with a set of unknown parameters to approximate the solution:

$$w(x,y) = \sum_{s=1}^{S} r_s \Phi_s(x,y) \tag{14}$$

The trial function is evaluated using the polynomial expansion:

$$\Phi(x,y) = \sum_{m=0}^{M} \sum_{n=0}^{N} a_{mn} x^n y^m \tag{15}$$

Having solved all the boundary conditions, we obtainthe values of the coefficients $a_{ij}$. Substituting the values of the coefficients in the general polynomial, we end up with trial function (14). The residual function R is determined as:

$$R(x,y) = \nabla^4 w(x,y) + pM + \omega^2 \hat{M} \text{ for CPT or more complicated for FSDT and TSDT} \tag{16}$$

For the residual function (16) the coefficients $r_i$ in Equation (14) are derived with the use of the Bubnov-Galerkin method.

## 4. Numerical Example

Let us consider a triangular plate clamped on all sides. For such boundary conditions, the expansion (15) is symmetric with respect to the line y = x and m and n should start from 2.

For plates made of FGM the pressure multiplier M (Figure 3a) reduces the value of the maximal displacement as the thickness t/L ratio increases. The reduction factor is higher for HSDT and it increases with more accurate approximations of deformations.

Let us note that the first mode of natural vibrations is identical to these plotted in Figures 4 and 5 (see Equation (13)). The comparison of natural modes is presented in Figure 6. The results are almost identical for those derived for isotropic plates.

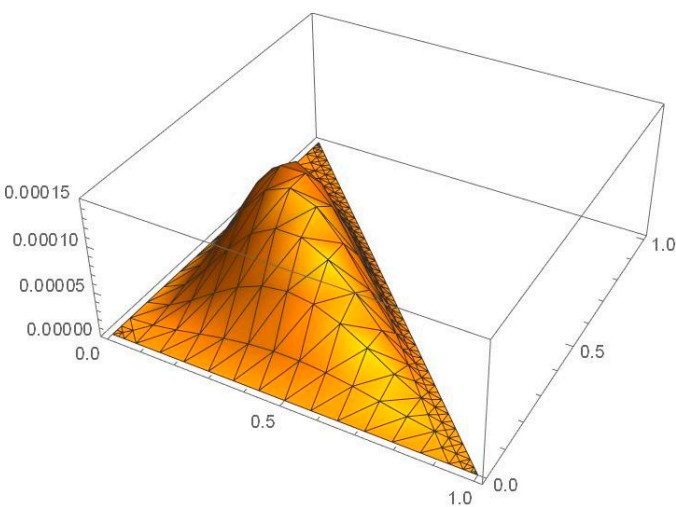

**Figure 4.** 3D plot of clamped plate normal deflection (C-C-C); the flexural rigidity $p/D$ = 1.

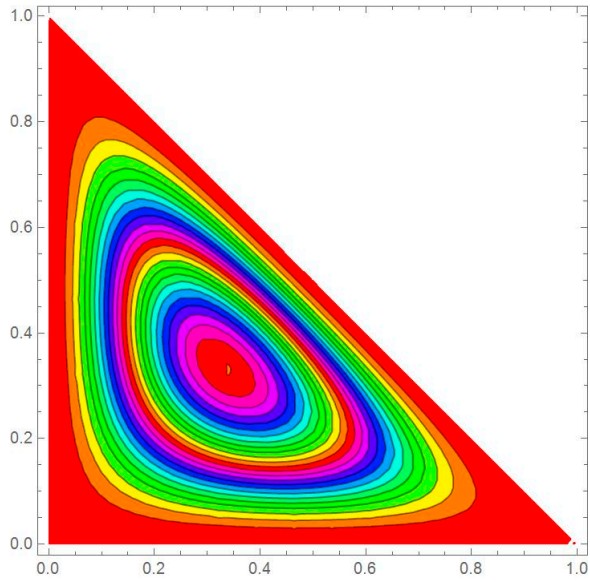

**Figure 5.** Contour plot of plate normal deflection for (C-C-C) plates; the flexural rigidity $p/D^t$ = 1.

| Boundary Conditions | First Mode | Second Mode | Third Mode | Fourth Mode | Fifth Mode |
|---|---|---|---|---|---|
| CCC | | | | | |
| CFC | | | | | |
| FCF | | | | | |

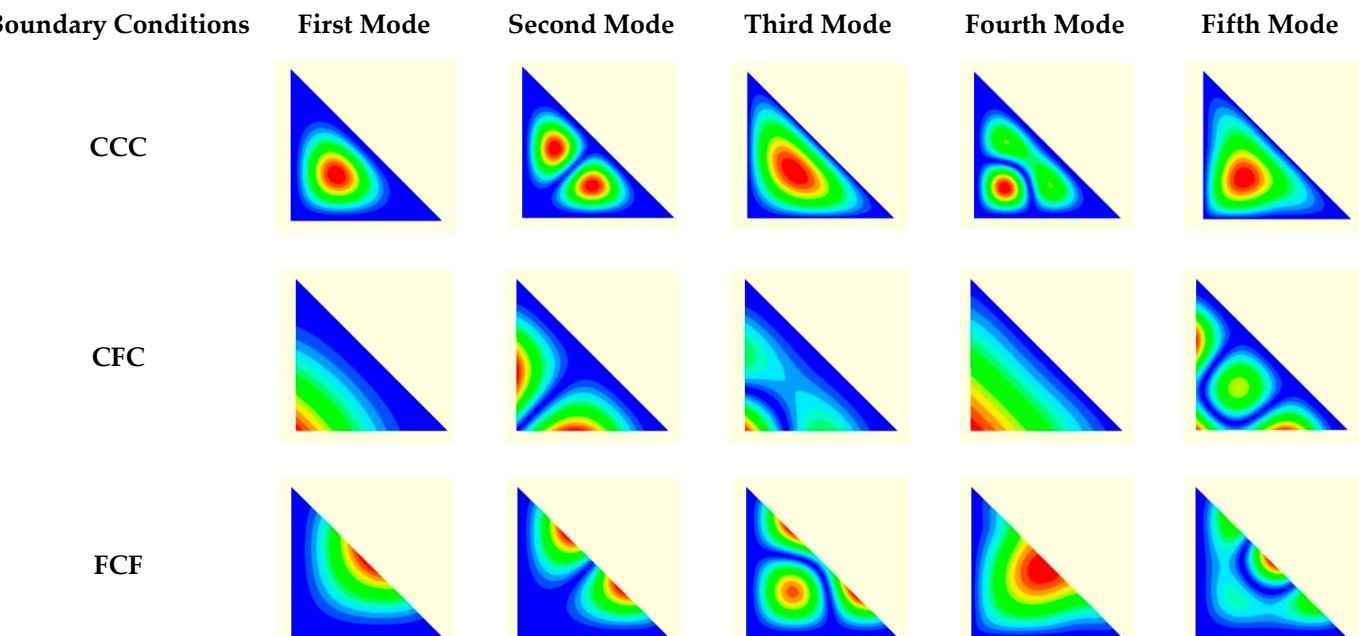

**Figure 6.** Modes of natural frequencies for different boundary conditions.

The natural frequency reduction factor is a function of the coefficient n and natural frequencies can be higher or lower than for isotropic plates (Figure 3b) similarly as for rectangular plates [2,7].

The validity of the analytical model was compared with the FE analysis for isotropic plates. The triangular plate deflections and natural vibrations can be evaluated with the use of finite element package NISA II in Table 2. Two types of finite elements were used, namely NKTP 20 (three kinematic variables) for thin-walledstructures (CPT) and NKTP 32 (five kinematic variables (FSDT and TSDT)). The agreement of results between analytical and FE analysis is very good. For moderately thick plates (t/L ≤ 0.1), the influence of transverse shear effects is less significant (see also Figure 6).

**Table 2.** Comparison of analytical and FE results for different 2D models of triangular clamped isotropic plates (t/L = 0.1, $D^t/p = 1$, $\rho^t/D^t = 1$).

| 2D Model | Normal Dimensionless Maximal Displacement– $w_{max}E^tt^3/[12p(1-\nu^2)]$ | | Dimensionless Eigenfrequency $\omega^2\sqrt{\rho^t/D^t}$–First Mode | |
|---|---|---|---|---|
| | **Analytical** | **FE (Nisa II)** | **Analytical** | **FE (Nisa II)** |
| CPT | $9.516 \times 10^{-5}$ | $9.744 \times 10^{-5}$ NKTP 20 | 89.4 | 95.4 NKTP 20 |
| FSDT | $9.464 \times 10^{-5}$ | $9.681 \times 10^{-5}$ NKTP32 | 86.3 | 93.2 NKTP 32 |
| TSDT | $9.412 \times 10^{-5}$ | $9.594 \times 10^{-5}$ NKTP32 | 85.1 | 90.4 NKTP 32 |

Both the results shown in Table 2 and in Figure 7 demonstrate that transverse shear effects reduce the values of maximal deformations and natural frequencies. It is in a good agreement (qualitative not quantitative) with the results obtained in refs [17,20].

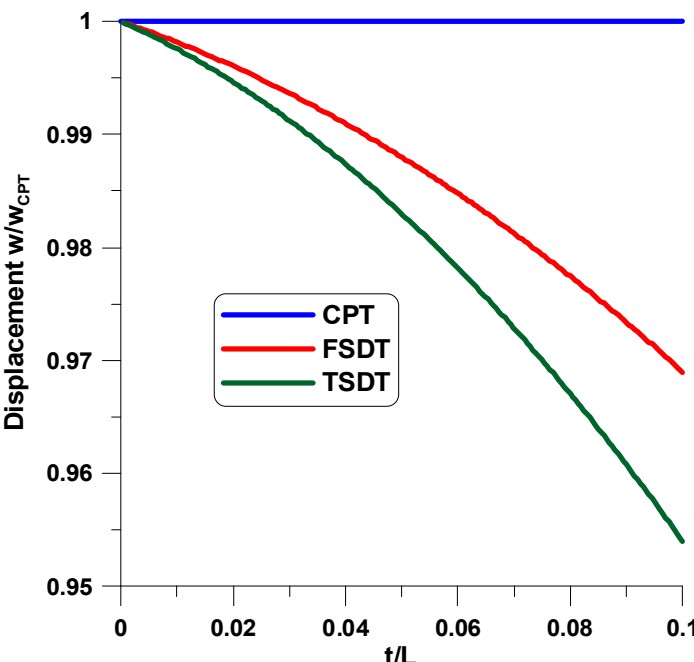

**Figure 7.** Variations of the clamped plate maximal deflections for different approximations of deformations.

## 5. Concluding Remarks

In the present paper, triangular moderately thick plates made of functionally graded porous materials are considered. The proposed model is based on three variants of 2D approximations of plate deformations (i.e., classical theory, first order shear theory, and third order shear theory). The problem deals with deformations and free vibration analysis. It is solved with the use of the weighting residuals and the Bubnov-Galerkin method. The validity and correctness of the results are verified by the comparison with the FE results derived with the use of NISA II package for isotropic structures.

The asymmetry of material characteristics of FG materials can be taken into account by the proposed two multiplication factors evaluated for classical plate theories.

**Funding:** This research received no external funding.

**Conflicts of Interest:** The authors declare no conflict of interest.

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
