# Peer review of "Triangular Functionally Graded Porous Moderately Thick Plates—Deformations and Free Vibrations"

_jcs, doi:10.3390/jcs5100276_

Round 1

Reviewer 1 Report

Based on the application of the weighted residuals and the Bubnov-Galerkin method, this paper presents a numerical method for simulating deformations and free vibrations of triangular functionally graded porous thick plates. Detailed comments and suggestions are given below.

  1. It would be useful to give more examples with the triangular functionally graded porous thick plates subjected to different boundary conditions.
  2. It would be better to investigate the effect of the porosity on its deformations and free vibrations.
  3.  In the introduction part, the author needs to present the readers' big pictures on the recent development of the novel numerical methods for the deformations and free vibrations of functionally graded thick plates. Regarding this aspect, some of the related published papers are listed as follows: a) Generalized Finite Difference Method for Plate Bending Analysis of Functionally Graded Materials[J]. Mathematics, 2020, 8(11): 1940. b) A simple-FSDT-based isogeometric method for piezoelectric functionally graded plates[J]. Mathematics, 2020, 8(12): 2177.

Author Response

They are included in the file

Reviewer 2 Report

In this paper, the triangular moderately thick plates made of functionally graded porous materials are considered. The proposed model is based on three variants of 2D approximations of plate deformations, i.e. classical theory, first order shear theory and third order shear theory. The problem deals with deformations and free vibration analysis. It is solved with the use of the weighting residuals and the Bubnov-Galerkin method. The validity and correctness of the results are verified by the comparison with the FE results derived with the use of NISA II package. In my option, this paper is recommended after a major revision as follows:

  1. In the introduction, the author simply lists several literatures, and does not give the advantages and disadvantages of different methods and the innovation of this paper in detail.
  2. Some formulas in this article are not standardized.
  3. The authors only given the comparison of analytical and FE results for different 2D models of triangular clamped isotropic plates. The authors should compare the results for the functionally graded porous plate with those by other methods.
  4. The chart in the paper is not clear.

Author Response

They are included in the filr

Round 2

Reviewer 2 Report

The authors have adopted the comment from the reviewers in my option. Therefore, I suggest acceptance for publication.